# MoDA: Multi-modal Diffusion Architecture for Talking Head Generation

## Abstract

Talking head generation with arbitrary identities and speech audio remains a crucial problem in the realm of the virtual metaverse. Despite progress, current methods still struggle to synthesize diverse facial expressions and natural head movements while generating synchronized lip movements with the audio. The main challenge is stylistic discrepancies between speech audio, individual identity, and portrait dynamics. To address the challenge of inter-modal inconsistency, we introduce MoDA, a multi-modal diffusion architecture with two well-designed technologies. First, MoDA explicitly models the interaction among motion, audio, and auxiliary conditions, enhancing overall facial expressions and head dynamics. In addition, a coarse-to-fine fusion strategy is employed to progressively integrate different conditions, ensuring effective feature fusion. Experimental results demonstrate that MoDA improves video diversity, realism, and efficiency, making it suitable for real-world applications.

## 1 Introduction

Talking head generation aims to create a photorealistic, speaking portrait from a single image, guided by audio and other modalities. Combined with the generative adversarial network (GAN) (Goodfellow et al., 2014) and diffusion model (Sohl-Dickstein et al., 2015), recent methods demonstrate widespread potential applications, such as immersive telepresence, and virtual characters.

Diffusion models have recently marked a significant advancement in generative modeling, enabling the creation of highly diverse videos. Early diffusion-based methods Cui et al. (2024a); Tian et al. (2024); Jiang et al. (2024); Xu et al. (2024a) generate the final video directly from the audio input. Although trainable from end to end, methods like Hallo2 Cui et al. (2024a) remain two major limitations persist, as shown in Fig. 1: 1) Inefficient inference process and visual artifacts. 2) Unnatural facial expressions and head movements with precise lip-sync. Recently, two-stage methods Xu et al. (2024b); Cao et al. (2024); Li et al. (2024) have simplified the diffusion process by bypassing complex variational auto-encoder (VAE) decoding. Methods like VASA-1 Xu et al. (2024b) first use the diffusion model to generate intermediate motion representations from audio, and then use a separate rendering network to synthesize the final video. However, the final video quality is heavily dependent on the accuracy of these intermediate representations. Thus, these methods still struggle to achieve natural facial dynamics with precise lip-sync due to suboptimal predictions.

Looking into the aforementioned issues, we argue that their root cause is the stylistic discrepancies between speech audio, individual identity, and portrait dynamics. These methods typically concatenate multiple conditions to form a mixed representation, which is then fed into a cross-attention mechanism where information flows only from this mixed modality to motion. This design introduces a learning bias, causing the model to focus only on the most shallow feature cues while neglecting the intricate relationships between the modalities. Consequently, it fails to handle more complex or conflicting scenarios. As in the ablation study, this limitation leads directly to inconsistent motion sequences when the model is conditioned on arbitrary identity input.

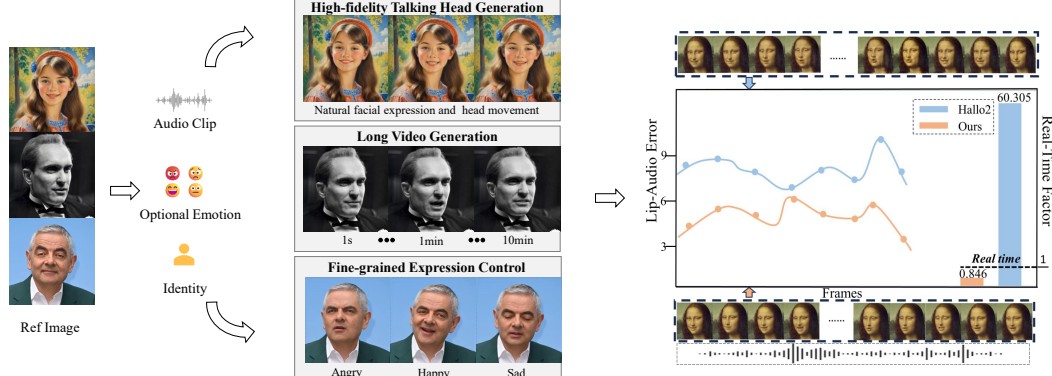

Figure 1: Beyond precise lip-sync, MoDA demonstrates strong capabilities in generating natural facial expressions and head movements. Real-Time Factor (RTF) measures the ratio of inference time to output duration.

To address the challenge of inter-modal inconsistency, we propose MoDA, a novel framework designed for synergistic talking head generation. MoDA begins by operating within a joint parameter space that bridges motion generation and neural rendering, encompassing a disentangled motion-appearance space Guo et al. (2024) along with audio, emotion, and identity. This space has a dimensionality that is an order of magnitude lower than traditional VAE spaces, which dramatically reduces the complexity of multi-modal fusion. Moreover, by incorporating optional conditions like identity and emotion, MoDA makes the generative modeling of complex distribution more tractable and increases fine-grained control over the generative process.

MoDA is guided by two core principles designed to address these inconsistencies: 1) We draw inspiration from recent lip-to-speech tasks Varshney et al. (2022); Prajwal et al. (2020a), where visual information can provide additional context to complement the audio. As shown in Fig. 2, MoDA introduces the Multi-modal Diffusion Transformer (MMDiT) Esser et al. (2024), equipped with rectified flow Liu et al. (2022), as the framework to facilitate multi-modal fusion. In this design, MoDA can dynamically adapt audio features based on motion, identity, and emotion, thereby improving the accuracy of motion generation. 2) To systematically integrate these modalities based on semantic information, MoDA implements a coarse-to-fine fusion strategy. Initially, the model uses separate weights to capture the unique characteristics of each modality. In the intermediate stage, a unified representational space is introduced for semantically linked modalities like audio, emotion, and identity to form a unified motion command. Finally, all modalities are integrated into a unified representation space, allowing holistic fusion. To further encourage precise lip-sync while maintaining motion diversity, MoDA provides an optional Adaptive Lip-motion sync Expert (ALSE), which can be integrated during training. The contributions of MoDA can be summarized as follows:

- This paper proposes MoDA, a novel multi-modal diffusion framework that generates high-fidelity talking head videos from an image, audio, and additional modalities.

- A coarse-to-fine fusion strategy is designed to progressively integrate noisy motion with audio and other modalities, enabling effective multi-modal fusion.

- Extensive evaluations on public datasets demonstrate that our method outperforms contemporary alternatives in visual quality and quantitative metrics.

## 2 RELATED WORK

### 2.1 DISENTANGLED FACE REPRESENTATION

Recent research on disentangled facial representation learning has explored methods using sparse keypoints Siarohin et al. (2019) or 3D Morphable Models (3DMM) Blanz & Vetter (1999); Li et al. (2017) to model facial dynamics. The 3DMM projects the 3D head shape into low-dimensional PCA spaces, allowing manipulation of attributes like identity, pose, and expression via linear blend

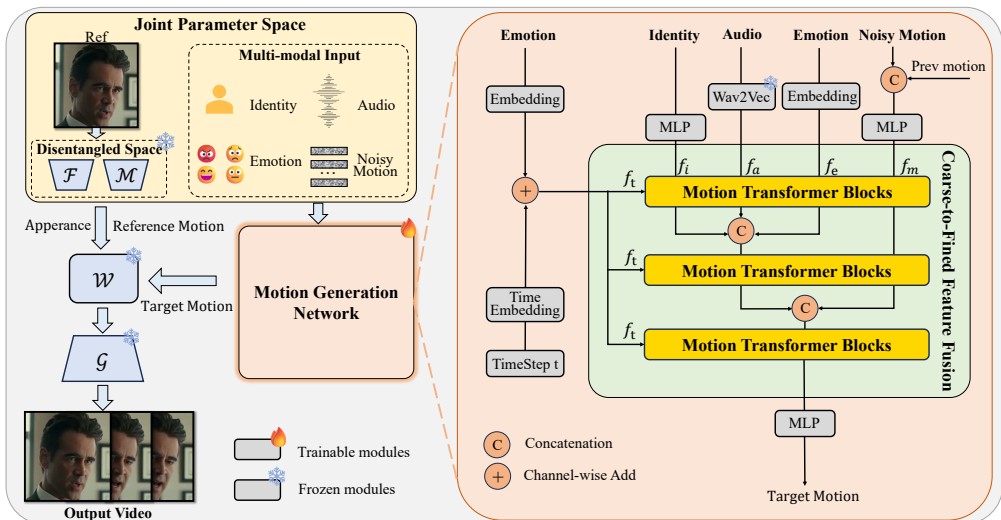

Figure 2: Overall architecture of the proposed MoDA and illustration of our Motion Generation Network. The appearance extractor $\mathcal{F}$, motion extractor $\mathcal{M}$, warping module $\mathcal{W}$, and decoder $\mathcal{G}$ are frozen. A motion feature generation model based on a Diffusion Transformer is then trained to generate motion features.

skinning. However, these methods may suffer from reconstruction inaccuracies or difficulties in decoupling facial attributes. Recent learning-based methods, such as Face Vid2Vid Wang et al. (2021), LivePortrait Guo et al. (2024), and MegaPortraits Drobyshev et al. (2022), use non-linear parameter spaces to improve the disentanglement of facial representation. These approaches capture more detailed facial expressions and offer greater flexibility in animation.

## 2.2 AUDIO-DRIVEN TALKING HEAD GENERATION

In audio-driven digital human technology, one-shot methods have gained attention for generating dynamic avatars from a single image. These methods are classified into single-stage and two-stage audio-to-video generation. Single-stage methods Jiang et al. (2024); Cui et al. (2024a); Prajwal et al. (2020b); Shi et al. (2024); Xu et al. (2024a); Guan et al. (2023); Cui et al. (2024b) map audio features directly to video frames. In contrast, two-stage methods Li et al. (2024); Xu et al. (2024b); Cao et al. (2024); Sun et al. (2023); Zhang et al. (2023) use intermediate representations like motion sequences or keypoints.

Early single-stage methods Guan et al. (2023); Prajwal et al. (2020b); Suwajanakorn et al. (2017) focused on lip-sync accuracy using GANs. Recent advances Xu et al. (2024a); Shi et al. (2024); Cui et al. (2024a); Tian et al. (2024); Chen et al. (2024) incorporated diffusion-based approaches, mapping audio to diverse facial expressions and head movements. However, these methods face challenges like high computational overhead and low inference efficiency due to denoising in the VAE space, where appearance and motion are entangled.

Two-stage methods address these limitations by using a disentangled facial space as an intermediate representation. Early methods Ye et al. (2023; 2022) utilized landmarks or 3DMM for motion synthesis. Recently, VASA-1 Xu et al. (2024b), Ditto Li et al. (2024), and JoyVASA Cao et al. (2024) shifted to implicit facial representations Wang et al. (2021); Drobyshev et al. (2022); Guo et al. (2024), employing DiT-based models for audio-to-motion mapping, resulting in more expressive video synthesis. However, relying solely on cross-attention for lip-sync generation limits the diversity and expressiveness by neglecting rich multi-modal interactions and deep-level information within the input signals.

## 3 METHOD

### 3.1 PRELIMINARIES

Denoising Diffusion Probabilistic Models (DDPMs) Song et al. (2020) have emerged as a powerful framework for generative modeling by formulating the data generation process as an iterative denoising procedure. In the forward diffusion process, Gaussian noise $\epsilon$ is gradually introduced into the data distribution across $T$ discrete timesteps, producing noisy latent features: $z_t = \sqrt{\alpha_t}z_0 + \sqrt{1-\alpha_t}\epsilon$, where $\alpha_t$ represents a variance schedule that determines the noise level at each timestep, and $z_0$ is the raw data. The model is trained to reverse this process by taking the noisy latent representation $z_t$ as input and estimating the added noise $\epsilon$. The training objective is defined as: $\mathcal{L} = \mathbb{E}_{z_t,c,\epsilon\sim\mathcal{N}(0,1),t}\left[||\epsilon - \epsilon_\theta(z_t,t,c)||_2^2\right]$, where $\epsilon_\theta$ denotes the noise prediction generated by the model, and $c$ represents additional conditioning signals, such as audio or motion frames, which are particularly relevant in the generation of talking videos. Recently, Stable Diffusion 3 (SD3) Esser et al. (2024) has advanced the paradigm by introducing Rectified Flow Pooladian et al. (2023); Liu et al. (2022) that optimizes the objective of traditional DDPMs:

$$\mathcal{L} = \mathbb{E}_{z_t,c,\epsilon\sim\mathcal{N}(0,1),t}\left[||(\epsilon - z_0) - v_\theta(z_t,t,c)||_2^2\right], \tag{1}$$

where $z_t = (1-t)z_0 + t\epsilon$, and $v_\theta$ denotes the velocity field. After the rectified flow training is completed, the transition from $\epsilon$ to $z_0$ can be formulated using the numerical integration of an ordinary differential equation (ODE):

$$z_{t-\frac{1}{N}} = z_t + \frac{1}{N}v_\theta(z_t,t,c), \tag{2}$$

where $N$ denotes the discretization steps of the interval $[0,1]$. The piecewise linear denoising process improves training stability. Furthermore, since our audio-to-motion task does not involve complex pixel information, this approach is particularly well-suited to our needs. Given these advantages, we adopt the rectified flow for training.

### 3.2 MODEL ARCHITECTURE

Instead of simply concatenating these conditioning signals, we enhance the intrinsic characteristics and emotional nuances of the audio representation by treating external emotion and identity cues as "catalysts". This approach balances identity and emotion between audio and speakers, enabling more natural lip control in real-world scenarios.

#### 3.2.1 JOINT PARAMETER SPACE

As shown in Fig. 2, we incorporate the existing facial re-enactment framework Guo et al. (2024) to extract disentangled facial representations. Specifically, the motion extractor $\mathcal{M}$ yields expression deformations $\delta$, head pose parameters $(R,t)$, the canonical keypoints of the source image $x_c$, and a scaling factor $S$. The motion representation $(R_s, \delta_s, t_s, S_s) \in \mathbb{R}^{70}$, serves as an identity-agnostic representation of the source input and is used to train MoDA to predict $(\hat{R}, \hat{\delta}, \hat{t}, \hat{S})$ given audio input.

$$\begin{aligned} x_s &= S_s \cdot (x_c R_s + \delta_s) + t_s \quad, \\ \hat{x} &= \hat{S} \cdot (x_c \hat{R} + \hat{\delta}) + \hat{t} \qquad. \end{aligned} \tag{3}$$

Subsequently, the warping field estimator $\mathcal{W}$ computes a field from $x_s$ and $\hat{x}$ to deform the 3D features $f_s$, which are then passed to the generator $\mathcal{G}$ to synthesize the target image. The audio features $f_a$ are extracted using the wav2vec Schneider et al. (2019) encoder. To maintain consistency in the identity feature space across various scenarios, we use the canonical keypoints of the source image $x_c$ as identity information and generate identity features $f_i$. For facial emotion signals, we use a visual emotion classifier Savchenko (2022) to extract the speaker's emotional labels and encode them into corresponding features $f_e$. The motion features from the first frame of each clip are used as previous motion to ensure inter-frame continuity and generate noisy motion features $f_m$. Inspired by EMO2 Tian et al. (2025), emotion features $f_e$ are added to timestep embeddings to generate timestep features $f_t$, which are injected into each motion transformer block by adaptive layer normalization (AdaLN) Peebles & Xie (2022). AdaLN is used to prevent the degradation of emotion features during the joint-attention operation, ensuring that emotional cues are preserved throughout the fusion. By integrating these conditioning signals, our model effectively generates realistic and temporally consistent motion features.

### 3.2.2 MOTION TRANSFORMER BLOCKS

As shown in Fig. 3 (c), the block consists of three main components: Modality-specific paths, Joint-attention and Rotational Position Encoding (RoPE) Yang et al. (2024). Modality-specific paths are designed to distinguish representations from different modalities. Specifically, each modality is equipped with its own adaLN and a modulation mechanism Peebles & Xie (2022) to improve the conditional generation capabilities of the model. Joint-attention is employed to interact across modalities. All modalities are first projected onto their respective query (Q), key (K), and value (V) representations, which are then concatenated in order along the sequence dimension. The combined sequence is processed through an attention operation, after which the attended features are split back into their respective modalities in the original order. To enhance temporal alignment between noisy motion and other conditional features, we adopt RoPE instead of the absolute positional encoding used in MMDiT. Specifically, we first expand the expression and identity features to match the sequence length of the noisy motion and audio features. We then apply aligned RoPE across all modalities, which facilitates better temporal sync and more consistent feature representations.

### 3.2.3 COARSE-TO-FINE FEATURE FUSION

Although assigning modality-specific paths with separate QKV projections and FeedForward Networks (FFNs) in the attention mechanism can enhance multi-modal information, this design often overlooks the inherent semantic commonalities shared across modalities. Independently learned weights hinder effective multi-modal feature fusion and introduce redundant parameters, potentially causing inconsistency issues. This issue is particularly pronounced in tasks like audio-to-motion generation, where no pixel-level inputs are involved, and the semantic gap between modalities is relatively small. In such cases, maintaining entirely separate parameterizations fails to provide meaningful benefits and instead leads to duplicated representations. To enhance the

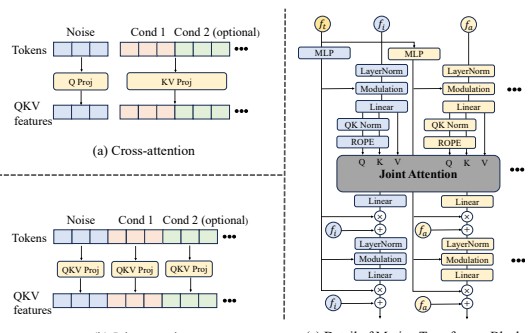

Figure 3: The detail of the Motion Transformer Block. (a) Cross-attention uses noise as the query and conditions as key and value. (b) Joint-attention projects noise and conditions separately and concatenates them before attention.

model's capacity for multi-modal understanding, we introduce a Coarse-to-Fine Feature Fusion strategy. This fusion strategy progressively integrates multi-modal features while reducing unnecessary parameters, thereby enhancing the multi-modal understanding and training stability. As illustrated in Fig. 2, the architecture progresses through three stages: a four-stream, two-stream, and single-stream process. In the four-stream stage, each modality is processed independently to learn modality-specific representations, facilitating early-stage feature differentiation. In the two-stream stage, the audio stream is concatenated with emotion and identity features to form a merged stream that shares weights. This design encourages a balanced integration of emotional and identity cues from both the audio and the reference image. Finally, all modalities are unified into a single representation stream to allow deeper fusion and enhance generative expressiveness.

### 3.2.4 LOSS FUNCTION

Our loss function utilizes the rectified flow loss and Eq. 1 is rewritten as:

$$\mathcal{L}_{RF} = \mathbb{E}_{z_t, c, \epsilon \sim \mathcal{N}(0,1), t} \left[ ||x - v_\theta(z_t, t, c)||_2^2 \right], \tag{4}$$

where $x$ represents $\epsilon - z_0$, $v_\theta$ denotes the velocity field. The velocity loss $L_{vel}$ is introduced to encourage improved temporal consistency:

$$\mathcal{L}_{vel} = ||x' - m'||_2^2 + ||x'' - m''||_2^2, \tag{5}$$

where $m$ denotes the output of $v_\theta(z_t, t, c)$, and $m''$. $m''$ denotes the first-order and second-order derivatives of $m$. To further enforce lip-sync accuracy, we design the ALSE pretrained in audio and motion features and compute the loss $\mathcal{L}_{ALSE}$. It is worth noting that supervision from the pretrained

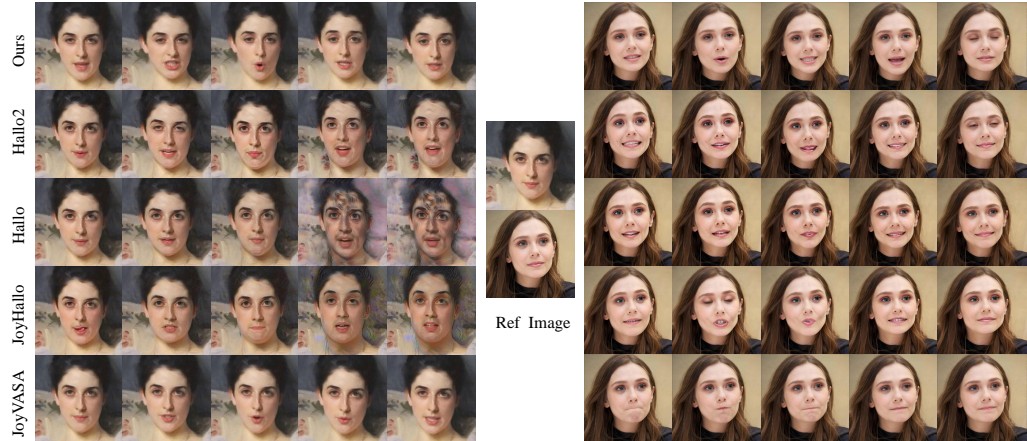

Figure 4: Qualitative Comparisons with State-of-the-Art Method. As single-frame images cannot fully represent sync, naturalness, and stability, we provide complete video comparisons in the supplementary materials.

Table 1: Comparison with existing methods on the HDTF and CelebV-HQ test sets. ↑ Higher is better. ↓ Lower is better. Best results are in **bold**, second-best are underlined.

| Dataset | Method | FVD ↓ | FID ↓ | F-SIM ↑ | Sync-C ↑ | Sync-D ↓ | Smo(%) ↑ |
|---------|--------|-------|-------|---------|----------|----------|----------|
| **HDTF** | GT | - | - | 0.860 | 7.267 | 7.586 | 0.9959 |
| | EchoMimic | 207.987 | 29.633 | 0.887 | 2.744 | 11.805 | 0.9939 |
| | JoyHallo | 256.226 | 44.842 | 0.852 | 7.360 | 7.984 | 0.9944 |
| | Hallo | 216.573 | 34.350 | 0.878 | 7.087 | 7.941 | 0.9950 |
| | Hallo2 | 229.806 | 34.426 | 0.871 | 7.102 | 7.976 | 0.9951 |
| | Ditto | 243.491 | 32.200 | 0.943 | 6.102 | 8.790 | 0.9970 |
| | JoyVASA | 229.634 | 32.584 | **0.953** | 5.255 | 9.600 | 0.9968 |
| | **Ours (sync)** | 191.292 | 30.449 | 0.925 | **8.183** | **7.065** | 0.9970 |
| | **Ours** | **174.622** | **28.182** | 0.927 | 7.369 | 7.744 | **0.9971** |
| **CelebV-HQ** | GT | - | - | 0.861 | 5.837 | 7.989 | 0.9964 |
| | EchoMimic | 258.451 | 47.169 | 0.837 | 2.610 | 11.216 | 0.9946 |
| | JoyHallo | 282.081 | 57.247 | 0.813 | 6.041 | 8.418 | 0.9945 |
| | Hallo | 245.101 | 44.411 | 0.851 | 5.629 | 8.384 | 0.9952 |
| | Hallo2 | 242.352 | 46.615 | 0.851 | 5.671 | 8.397 | 0.9953 |
| | Ditto | 302.525 | 46.996 | 0.915 | 4.681 | 9.280 | **0.9973** |
| | JoyVASA | 271.231 | 44.574 | **0.918** | 5.171 | 8.632 | 0.9971 |
| | **Ours (sync)** | **205.307** | 44.201 | 0.916 | **6.552** | **7.635** | 0.9972 |
| | **Ours** | 205.442 | **44.071** | 0.913 | 5.878 | 8.135 | 0.9972 |

ALSE is optional and can be applied as needed. In summary, the final loss can be expressed as follows:

$$\mathcal{L} = \mathcal{L}_{RF} + \mathcal{L}_{vel} + \lambda_{sync} \cdot \mathcal{L}_{ALSE} \quad , \tag{6}$$

$$\lambda_{sync} = \begin{cases} 1, & \text{if } \mathcal{L}_{ALSE} < \tau \\ 0, & \text{otherwise} \end{cases} \quad , \tag{7}$$

where $\tau$ is set to 0.4, controlling sync loss activation to ensure lip-motion supervision is applied only after establishing basic motion diversity. Further details about the ALSE can be found in the supplementary materials.

### 3.3 REALTIME INFERENCE

The real-time conversational scenario is enabled by low-latency motion generation. We align audio features with the video frame rate and segment the audio into continuous 100-frame chunks for streaming generation. Additionally, leveraging low-dimensional intermediate representations and

the rectified flow, we reduce DiT inference denoising steps from 50 to 10, yet achieve even higher quality. For detailed comparison, please refer to the supplementary material.

### 3.3.1 CLASSIFIER-FREE GUIDANCE (CFG)

In the training stage, we randomly assign each of the input conditions, and during inference, we perform the following:

$$\hat{z}^0 = (1 + \sum_{c \in C} \lambda_c) \cdot v_\theta(z^t, t, C) - \sum_{c \in C} \lambda_c \cdot v_\theta(z^t, t, C|c = \emptyset), \tag{8}$$

where $\lambda_c$ is the CFG scale of condition c. $C|c = \emptyset$ denotes that the condition c is $\emptyset$.

## 4 EXPERIMENTS

### 4.1 EXPERIMENT SETTINGS

#### 4.1.1 DATASET AND METRICS

Our motion generation model is primarily trained on three publicly available datasets: HDTF Zhang et al. (2021), CelebV-Text Yu et al. (2023), and MEAD Wang et al. (2020). For evaluation, we conducted experiments on three distinct test sets. The first two are derived from public datasets, CelebV-HQ Zhu et al. (2022) and HDTF, each consisting of 50 randomly sampled clips ranging from 3 to 10 seconds in length. The third is an in-the-wild set with 20 diverse cases, including real individuals, animated characters, dynamic scenes, and complex headwear. Each sample is paired with audio that is speech, emotional dialogue, or singing. We utilize several evaluation metrics to assess the performance of the proposed method. The Fréchet Inception Distance (FID) Seitzer (2020) and the Fréchet Video Distance (FVD) Unterthiner et al. (2019) are used to assess the quality of the generated output, while the F-SIM Tian et al. (2024) measures facial similarity. In addition, Sync-C Chung & Zisserman (2017) and Sync-D Chung & Zisserman (2017) metrics are introduced to evaluate lip-sync between different methods. A temporal smoothness metric (Smo) Huang et al. (2024) is also utilized to monitor the continuity of generated motion.

#### 4.1.2 IMPLEMENTATION DETAILS

During training, we randomly sample 80-frame segments from video clips to train the motion generation model. The model is trained for approximately 500 epochs on 8 NVIDIA H20 GPUs with a batch size of 512, using the Adam optimizer with a learning rate of 1e-4. During training, we apply a dropout probability of 0.1 for each emotion condition, while the dropout probability for speech is set to 0.5. Furthermore, the model is structured with 3 four-stream blocks, 6 two-stream blocks, and 12 single-stream blocks.

Table 2: Comparison with methods on the wild test dataset.

| Method | F-SIM ↑ | Sync-C ↑ | Sync-D ↓ | RTF ↓ |
|---|---|---|---|---|
| EchoMimic | 0.870 | 2.292 | 12.130 | 48.657 |
| JoyHallo | 0.825 | 7.000 | 8.167 | 59.735 |
| Hallo | 0.848 | 6.051 | 8.730 | 59.190 |
| Hallo2 | 0.849 | 6.386 | 8.523 | 60.305 |
| Ditto | 0.923 | 6.107 | 9.040 | **0.792** |
| JoyVASA | **0.924** | 5.569 | 9.368 | 1.717 |
| **Ours (sync)** | 0.896 | **7.710** | **7.469** | 0.846 |
| **Ours** | 0.895 | 6.862 | 8.088 | 0.846 |

### 4.2 RESULTS AND ANALYSIS

We juxtapose the results of the proposed method against those of EchoMimic Chen et al. (2024), Ditto Li et al. (2024), JoyVASA Cao et al. (2024), JoyHallo Shi et al. (2024), Hallo Xu et al. (2024a), and Hallo2 Cui et al. (2024a), Ours and Ours (Sync) (MoDA with ALSE).

### 4.2.1 QUANTITATIVE COMPARISON

The quantitative results on the CelebV-HQ test and HDTF dataset are shown in Table 1. On the HDTF dataset, MoDA consistently outperforms all existing methods across all evaluation metrics. In particular, our method achieves the lowest FID and FVD scores, outperforming the second-best methods by 4.9 % and 16.0 %, respectively. This demonstrates the superiority of our method in terms of visual naturalness and the overall quality of the generated frames. The two-stage methods better preserve identity, as evidenced by their F-SIM scores. Notably, JoyVASA and Ditto achieve the highest and second-highest F-SIM, respectively—likely due to their relatively constrained head movements and facial expressions, which enhance frame-wise structural similarity. The correspondingly lower FVD scores further reflect these limitations. In contrast, MoDA effectively mitigates such issues, demonstrating that enhancing multi-modal fusion can lead to improved overall model performance.

On the CelebV-HQ test dataset, MoDA consistently outperforms the six baseline methods in all metrics except Sync-D and Smo, highlighting its robustness. Although MoDA slightly underperforms JoyHallo in sync confidence (Sync-C), it achieves a notable improvement in sync distance (Sync-D), along with further gains in motion smoothness. Tables 1 also report the results with and without ALSE. Introducing ALSE to supervise the sync between audio and keypoints leads to a significant improvement in lip-sync while maintaining high visual quality. Table 2 shows that our method outperforms existing approaches across multiple metrics on the wild test set with diverse identities. We also report the real-time factor (RTF), where RTF <1 indicates real-time capability. Ditto achieves the lowest RTF, benefiting from TensorRT acceleration. Our method also delivers competitive efficiency, demonstrating its suitability for real-world applications.

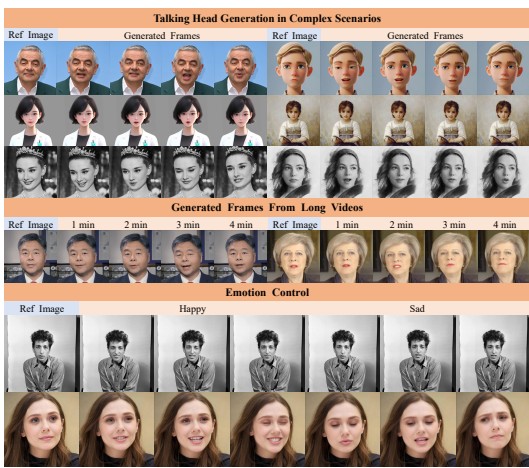

Figure 5: Generation results for portraits and audio in diverse styles are presented. We also demonstrate long-video inference and fine-grained control over facial expressions.

### 4.2.2 QUALITATIVE EVALUATION

As shown in Fig. 4, we selected two types of cases from the wild dataset for visual comparison. For each character, we generated videos using each method and selected frames from the same location for comparison. Analyzing the results, previous one-stage-based methods, including EchoMimic, Hallo2, Hallo, and JoyHallo, suffer from appearance blurring and unnatural expressions during temporal inference due to the strong entanglement between appearance and motion. In contrast, our proposed two-stage framework effectively mitigates these issues, ensuring high consistency in the generated details. Compared to the two-stage method JoyVASA and Ditto, MoDA generates richer expressions, better lip-sync, and more natural head movements, thanks to our multi-modal motion generation network, which effectively integrates deep information across different modalities.

### 4.2.3 VISUALIZATION RESULTS IN COMPLEX SCENARIOS

We further investigate MoDA's generation performance in complex scenarios. Specifically, for the visual modality, we utilize portraits of both real humans and animated characters, each paired randomly with various audio types, including speech, singing, recitation, and others. As shown in Fig. 5, our method demonstrates strong performance in various complex scenarios. In addition, it is capable of long-duration inference and fine-grained facial expression control.

### 4.3 ABLATION STUDY

For more ablation experiments and implementation details, see the Appendix.

### 4.3.1 CROSS-ATTENTION

We replace the MoDA with a Cross-Attention-Based Architecture (CABA) to perform multi-modal fusion for evaluation purposes. As shown in Table 3, this substitution results in a performance drop in all metrics. Moreover, as illustrated in Fig. 6, the subject exhibits mouth-closing failures when lacking deep multi-modal interaction. To investigate this, we conducted two experiments under the same settings used in the w/ CABA variant as show in Fig. 6: 1) Replacing the audio with the same image did not resolve the mouth closure issue (Replace audio). 2) Replacing the image with the same audio addressed the issue (Replace image). These results suggest that cross-attention fails to adapt audio features to different identity conditions. In contrast, MoDA enables dynamic adaptation of audio features based on contextual factors such as motion, identity, and emotion.

Table 3: Ablation study results on CelebV-HQ test sets.

| Method | FID $\downarrow$ | FVD $\downarrow$ | Sync-C $\uparrow$ | Sync-D $\downarrow$ |
|---|---|---|---|---|
| w/ CABA | 47.365 | 232.291 | 5.331 | 8.692 |
| w/o C2F | 44.548 | 221.631 | 5.535 | 8.465 |
| w/ MAF | 48.358 | 216.982 | 5.511 | 8.527 |
| Full Model | 44.071 | 205.442 | 5.878 | 8.135 |

### 4.3.2 COARSE-TO-FINED FEATURE FUSION

We replace the Coarse-to-Fine Feature Fusion with a fully four-stream architecture (w/o C2F). As shown in Table 3, this design leads to performance degradation, particularly in sync and motion quality, emphasizing the necessity of progressive feature fusion for effective cross-modal integration. These results indicate that independent learning of weights introduces substantial redundant parameters, resulting in inconsistency challenges (with 904M parameters in the w/o C2F variant).

In contrast, by gradually sharing weights, the proposed coarse-to-fine strategy significantly reduces the parameter count to 370M, leading to both greater efficiency and improved performance. To further investigate the effectiveness of the fusion strategy, we modified the original dual-stream design by creating an MAF variant that first merges motion with the auxiliary cues (emotion + identity) into one branch, while audio stays in the other. Table 3 also shows that MAF performs worse than even the w/o C2F baseline. Directly fusing the heterogeneous noisy motion and auxiliary condition features confuses the network. Our full model first merges audio with the auxiliary cues; Speech audio inherently carries emotion and identity signals, providing a natural bridge and delivering better accuracy and efficiency.

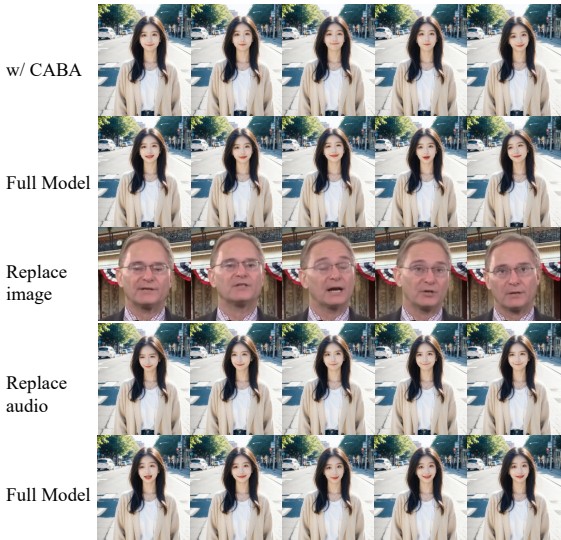

Figure 6: Ablation Study Visualizations.

## 5 CONCLUSION

We propose MoDA, a two-stage multi-modal diffusion framework for one-shot talking head generation. MoDA effectively leverages multi-modal information to map audio to motion sequences in an identity-agnostic latent space, which are then translated into video frames by a pre-trained face renderer. From a single portrait, MoDA produces high-quality, expressive, and controllable talking head videos, surpassing previous methods in quality, diversity, and naturalness with high efficiency.

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

## A  APPENDIX

### A.1  DETAILS OF ADAPTIVE LIP-MOTION SYNCHRONIZATION EXPERT

#### A.1.1  ARCHITECTURE

Our synchronization expert takes as input a sequence of $T_l$ consecutive lip keypoint frames and an audio feature clip of size $T_a \times D$, where $T_l$ and $T_a$ represent the number of frames for keypoints and audio, respectively, and $D$ denotes the dimensionality of Wav2Vec features Schneider et al. (2019). The expert aims to determine whether the input motion and audio are temporally aligned. As illustrated in Fig. 7, it consists of two parallel encoders for landmarks and audio, each built from a stack of 1D convolutions followed by batch normalization and ReLU activation. The training objective uses a cosine-similarity-based binary cross-entropy loss. Specifically, cosine similarity is computed between the keypoints embedding $l$ and the audio embedding $a$ to supervise synchronization.

$$\mathcal{L}_{sync} = CE\left(\frac{a \cdot l}{\max(\|a\|_2 \cdot \|l\|_2, \epsilon)}\right) \tag{9}$$

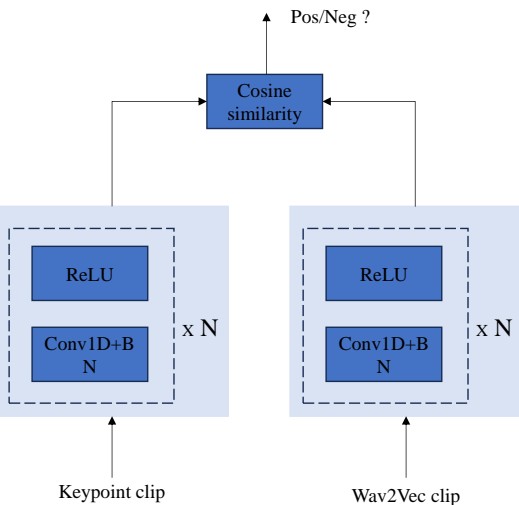

Figure 7: The structure of Adaptive Lip-motion Synchronization Expert

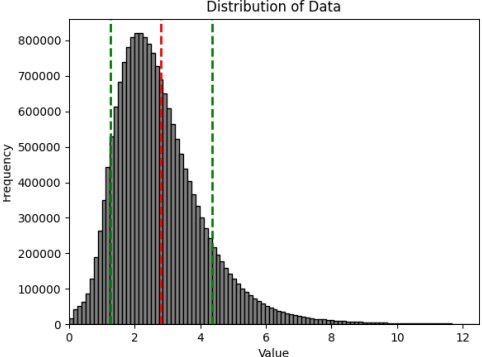

Figure 8: Distribution of motion intensities in the training dataset.

### A.1.2 TRAINING DETAILS

We train the Adaptive Lip-motion Synchronization Expert (ALSE) on a single NVIDIA H20 GPU, with a batch size of 512. The model is optimized using the Adam optimizer with an initial learning rate of $1 \times 10^{-4}$, which is decayed by a factor of 0.02. Training is performed for a total of 30,000 steps. The training dataset is kept consistent with that used for MoDA, with data distribution across different motion intensities illustrated in Fig. 8.

Table 4: Comparison of MoDA with different inference steps on the CelebV-HQ test set.

| Method | FVD ↑ | Sync-C ↑ | Sync-D ↓ | RTF ↓ |
|--------|-------|----------|----------|-------|
| Our-s10 | **205.442** | 5.804 | **8.182** | **0.846** |
| Our-s20 | 219.762 | **5.855** | 8.205 | 1.048 |
| Our-s30 | 213.908 | 5.751 | 8.295 | 1.259 |
| Our-s40 | 213.483 | 5.741 | 8.297 | 1.430 |
| Our-s50 | 214.182 | 5.757 | 8.346 | 1.635 |

Table 5: Single–step inference latency (seconds) for each module.

| Component | Latency |
|---|---|
| Pre-process | 0.1303 |
| Denoising Network | 0.2012 |
| Rendering | 6.3202 |

Table 6: User study with 20 participants (scores range from 1 to 5).

| Method | Lip Sync ↑ | Motion Diversity ↑ | ID Similarity ↑ |
|---|---|---|---|
| EchoMimic | 2.6 | 2.4 | 3.0 |
| JoyHallo | 3.3 | 2.8 | 2.9 |
| Hallo | 2.8 | 2.7 | 3.1 |
| Hallo2 | 3.2 | 3.1 | 3.4 |
| JoyVASA | 2.4 | 1.5 | 4.4 |
| Ditto | 3.1 | 2.9 | 4.5 |
| Ours | **4.5** | **4.2** | **4.1** |

## A.2 REAL-TIME INFERENCE

All experiments were run on one NVIDIA H20 GPU, whose inference speed is comparable to that of an NVIDIA RTX 3090. Relative to VAE-based approaches, MoDA adopts a far more compact latent representation—70 channels, compared with Hallo's $64 \times 64 \times 4$ feature map. As a result, synthesizing 16 frames entails 11,599.69 GFLOPs for Hallo but only 19.53 GFLOPs for MoDA. In addition, we replace the standard DDPM sampler with rectified flow, cutting the number of denoising steps from 50 to 10 while attaining even higher visual quality (see Table 4). These results indicate that MoDA allows us to trim 80 % of the diffusion iterations without compromising visual quality and lip-sync accuracy. Consequently, MoDA delivers state-of-the-art realism at real-time inference speed. A fine-grained runtime profile for generating a 10-second video is reported in Table 5.

## A.3 USER STUDIES

We conducted a user study on the public HDTF dataset, where 20 participants rated the results of six methods on a five-point scale with respect to three key aspects: lip-sync quality, motion diversity, and identity similarity. As reported in Table 6, MoDA achieves the highest scores on all criteria, outperforming every competing approach.

## A.4 EMOTION CONTROL

It should be stressed that the perceived emotion in a talking-head sequence is closely coupled with the accompanying audio and can, in most cases, be inferred directly from it. We therefore treat the external emotion cue merely as a "catalyst" that enables MoDA to infer this affect more effectively from the speech signal. The cue is applied only when necessary, to gently amplify or attenuate the latent emotion; it is not designed to perform a full emotion transfer or to synthesize expressions that contradict the input soundtrack. To verify this claim, we reran inference with different emotion codes and measured both lip-sync accuracy and FVD. As shown in Table 7, altering the emotion code does not degrade the model's generation quality.

## A.5 ADDITIONAL ABLATIONS AND RESULTS

### A.5.1 CROSS-ATTENTION

For the cross-attention, we follow Ditto's design Li et al. (2024): the audio, identity, and emotion embeddings are concatenated along the channel dimension, processed by a four-layer MLP, and then used as the key–value inputs in a cross-attention operation with the noise latent. We also identify an additional weakness of the cross-attention baseline in side-view scenarios.

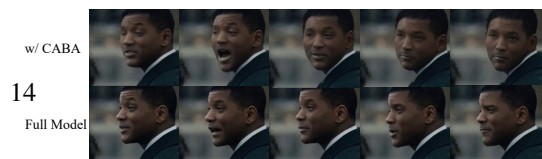

Table 7: Impact of forcing different emotion labels during inference on lip-sync accuracy and perceptual quality.

| Emotion | Sync-C ↑ | Sync-D ↓ | FVD ↓ |
|---------|----------|----------|-------|
| Anger | 5.885 | 8.160 | 207.325 |
| Contempt | 5.802 | 8.210 | 215.016 |
| Disgust | 5.797 | 8.220 | 220.178 |
| Fear | 5.855 | 8.131 | 203.160 |
| Happiness | 5.737 | 8.356 | 214.524 |
| Neutral | 5.804 | 8.182 | 206.250 |
| Sadness | 5.802 | 8.227 | 208.972 |
| Surprise | 5.844 | 8.197 | 212.952 |
| None (ours) | 5.878 | 8.135 | 205.442 |

Table 8: Ablation study results on major architectural choices.

| Method | FVD ↓ | Sync-C ↑ | Sync-D ↓ |
|--------|-------|----------|----------|
| w/ DDPMs | 220.291 | 5.433 | 8.832 |
| w/ RF-L1 | 203.371 | 5.543 | 8.656 |
| w/o E&I | 208.372 | 5.387 | 8.619 |
| w/ AdaLN | 216.871 | 5.442 | 8.441 |
| Full Model | 205.442 | 5.878 | 8.135 |

As shown in Fig. 9, the head generated with cross-attention rotates toward a frontal pose that does not match the side-view reference. This failure again reflects cross-attention's inability to resolve the inherent one-to-many ambiguity of talking-head generation, causing the model to gravitate toward the frontal orientations prevalent in the training data. By contrast, MoDA exploits the spatial cues in the reference frame to dynamically modulate the audio features, yielding videos whose head poses remain faithful to the intended side view and appear far more natural.

### A.5.2 RECTIFIED FLOW

We conduct ablation studies on the components of the loss function, as summarized in Table 8. Replacement of rectified flow with standard DDPMs leads to performance degradation across all metrics. Using an L1 loss (w/ RF-L1) improves visual quality (lower FVD), but reduces the accuracy of lip synchronization. Given the importance of synchronization in talking head generation, we adopt rectified flow with an L2 loss (Full Model) as the default setting.

### A.5.3 CONDITIONAL INJECTION METHOD

In our model, the conditions of identity and emotion are treated as two additional streams to balance the identity and emotional signals within the audio. To evaluate this design, we retain only the noise and audio streams, and compare two settings: one where emotion and identity conditions are injected via AdaLN ("w/ AdaLN") Peebles & Xie (2022), and another that omits the emotion and identity conditions, training MMDIT solely on the audio and noisy motion ("w/o E&I").

As shown in Table 8, when the AdaLN method is used, overall performance drops, suggesting that the extraction of identity and emotional cues from the audio is crucial to balance the inter-modal inconsistency.

Additionally, when identity and emotion conditions are not injected, we observe a significant decline in lip synchronization metrics. This suggests that, without these auxiliary conditions to act as catalysts, it is difficult to rely on the audio alone to generate consistent mouth shapes and natural movements. Interestingly, this variant attains a slightly lower (better) FVD than the AdaLN counterpart,

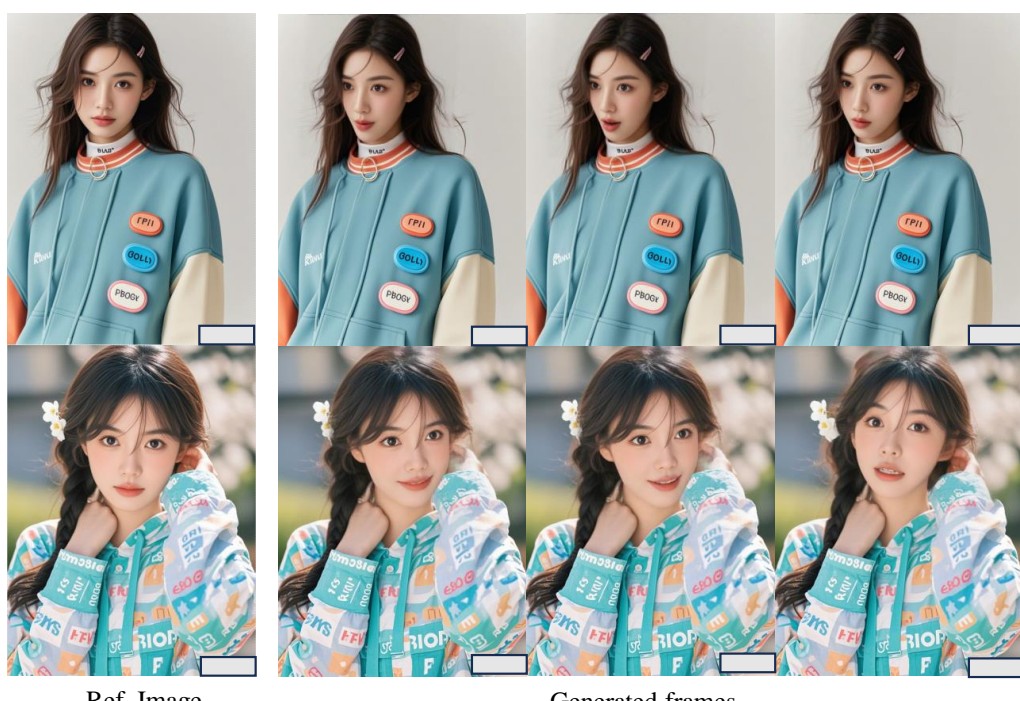

Ref  Image                    Generated frames

Figure 10: Examples illustrating the limitations.

implying that the audio stream already carries rich emotional and identity cues and that the AdaLN strategy cannot fully reconcile the inter-modal inconsistency. Moreover, regardless of whether additional conditions are injected, overall metrics remain superior to those of the cross-attention-based method, further demonstrating the importance of effective information interaction.

## A.6  LIMITATIONS

The main limitation of our framework stems from the first-stage model, where the Liveportrait generator struggles with large pose variations and complex head accessories. As shown in Fig. 10, these issues result in a noticeable decline in output quality, particularly under large pose variations and in the presence of complex head accessories. Significant pose changes often cause unnatural distortions, which are visually disturbing for users. Additionally, complex headwear can be misinterpreted as part of the background, leading to temporal inconsistencies and blurring between video frames.

