# OpenReview forum: "MoDA: Multi-modal Diffusion Architecture for Talking Head Generation"
_ICLR.cc/2026/Conference — ICLR 2026 Conference Withdrawn Submission_

### Official Review · Reviewer_JiHZ · 2025-10-22

**Soundness:** 2
**Presentation:** 2
**Contribution:** 2
**Rating:** 4
**Confidence:** 5

**Summary:**

This paper introduces MoDA (Multi-modal Diffusion Architecture), which generates realistic talking head videos from a single image, guided by audio and additional modalities such as emotion and identity. MoDA employs a multi-modal diffusion framework to map audio to motion sequences in an identity-agnostic latent space, which are then rendered into video frames. While results show that MoDA performs well across various test sets, the contribution remains unclear due to limited innovation and an ambiguous methodology.

**Strengths:**

- MoDA demonstrates improved performance over several existing methods (e.g., EchoMimic, JoyVASA) in terms of metrics like FVD, FID, and Sync.
- The framework delivers high efficiency, supporting real-time inference, which makes it suitable for practical applications.

**Weaknesses:**

- Many of MoDA’s methods and techniques rely on existing models, such as multi-modal fusion and audio-to-video mapping, without offering significant innovations or breakthroughs. The framework primarily combines established approaches.
- The methodology section is unclear, especially regarding the integration of multiple modalities (emotion, identity, audio) and how these components interact within the architecture. The technical details are underexplained, making it hard to understand the model’s workings.
- The flow of the paper is inconsistent, and it lacks sufficient context for the presented techniques.
- There is a need for more detailed ablation studies to demonstrate the impact of each component, and a clearer presentation of how these components contribute to the overall performance.
- Lacks comprehensive comparison with state-of-the-art methods such as VividTalk, Hallo3, and Hallo4.
- What is f_s in the framework? The explanation between Fig. 2 and the method is inconsistent, which makes it confusing.

**Questions:**

- The paper mentions that MoDA outperforms existing methods in certain metrics, but what makes MoDA fundamentally different in terms of theory and technique compared to methods like EchoMimic or JoyVASA?
- The methodology briefly mentions this feature fusion strategy, but the paper does not provide enough detail on why this particular strategy is superior and how it specifically contributes to improving video quality.
- The integration of audio, emotion, and identity features is mentioned, but the paper lacks a clear explanation of how these modalities interact and how this fusion enhances the overall performance.
- The paper includes some ablation experiments, but the analysis is not deep enough. How do each of the components, like "Motion Transformer Blocks" and "Joint Parameter Space," impact the model’s final performance? What are the contributions of individual components?

---

### Official Review · Reviewer_Bco2 · 2025-10-25

**Soundness:** 3
**Presentation:** 3
**Contribution:** 3
**Rating:** 8
**Confidence:** 4

**Summary:**

This paper introduces MoDA, a two-stage framework for generating high-fidelity talking heads. The proposed approach surpasses existing methods in visual realism and lip-sync accuracy while maintaining strong temporal consistency, enabling long-duration video generation.

**Strengths:**

(1)It explicitly models the interactions among motion, audio, and auxiliary conditions, effectively mitigating stylistic inconsistencies across modalities and significantly enhancing overall generation quality.

(2)The proposed Motion Transformer Block achieves efficient multi-modal feature fusion through modality-specific paths, adaptive normalization, and modulation mechanisms, leading to more natural and coherent talking head synthesis.

(3)The MoDA architecture balances high-quality generation with real-time performance.

**Weaknesses:**

(1)The experimental comparisons focus on Hallo2, without evaluating against more recent and advanced models such as Hallo3.
(2）Although MoDA is relatively fast, its training still requires substantial computational resources. Demonstrating its performance on consumer-grade GPUs would further validate its efficiency.

**Questions:**

（1）Could the authors clarify why the emotional effects appear to be limited? Is it because the emotion features rely on classifier-derived labels rather than being directly extracted from the audio, which might weaken the spontaneous alignment between emotion and audio?
（2）Could the authors clarify whether the outputs of the diffusion model represent relative motion or absolute motion, and whether one has been found to perform better than the other?

---

### Official Review · Reviewer_xVhE · 2025-10-25

**Soundness:** 2
**Presentation:** 2
**Contribution:** 2
**Rating:** 2
**Confidence:** 5

**Summary:**

This paper proposes MoDA, a multi-modal diffusion architecture designed for talking head generation from a single image, integrating motion, audio, emotion, and identity cues into a unified generation process. MoDA introduces a Multi-Modal Diffusion Transformer (MMDiT) equipped with rectified flow for more stable training and efficient inference. It also implements a coarse-to-fine (C2F) feature fusion strategy to progressively merge modality-specific features, reducing redundancy and improving semantic consistency. The model achieves better FVD/FID and lip-sync scores than existing methods such as Hallo2, Ditto, and JoyVASA across multiple benchmarks, while supporting real-time inference and controllable emotion-driven generation.

**Strengths:**

- Unified Multi-Modal Diffusion Framework : The paper introduces a well-structured MMDiT-based architecture that explicitly fuses motion, emotion, identity, and audio modalities. This design targets the persistent inter-modal inconsistency found in existing diffusion-based talking head methods.
- Coarse-to-Fine Feature Fusion : The proposed C2F mechanism effectively balances modality-specific processing and shared fusion, improving both parameter efficiency and motion consistency. The ablation study demonstrates performance drops without C2F (Table 3), supporting its design rationale.
- High Efficiency and Real-Time Capability : By leveraging rectified flow, MoDA reduces diffusion steps (50→10) while maintaining or improving quality, achieving near real-time inference (RTF ≈ 0.8). This makes it viable for real-world applications.
- Improved Quantitative Results : The model achieves lower FID/FVD and competitive lip-sync accuracy across HDTF, CelebV-HQ, and in-the-wild datasets. It outperforms baselines in most metrics, confirming that multi-modal conditioning helps mitigate over-smoothing and unnatural motion.
- Well-Analyzed Architectural Components : The inclusion of detailed ablation studies on cross-attention vs. joint-attention, AdaLN, and modality injection clarifies how each component contributes to performance.

**Weaknesses:**

- Modality Design Justification : The inclusion of emotion, identity, audio, and motion as four separate modalities seems ad hoc. Other plausible combinations (e.g., audio + identity, audio + expression) are not tested. The paper lacks a systematic ablation on modality selection or their individual contribution.
- Classifier-Free Guidance Reliance : Most conditional inputs except motion are used primarily as classifier-free guidance, which weakens the argument for deep joint modeling. This could limit the robustness and interpretability of the MMDiT’s fusion design, especially with small datasets.
- Emotion Conditioning Validation : The results in the supplementary section show emotion control limited to only two cases (happy, sad). It’s unclear whether emotion embedding truly influences finer-grained facial motion beyond simple label conditioning.
- Potential Overfitting : Given the limited data (mostly HDTF, MEAD, and CelebV-Text), the heavy MMDiT model may easily overfit to speaker-specific traits. The paper should report cross-speaker or unseen-identity generalization results.
- Inconsistent Visual Quality in Complex Scenes : The qualitative results reveal some unrealistic or unstable backgrounds in complex scenarios (Fig. 10). The method inherits limitations from LivePortrait-based rendering, which struggles with occlusions, large pose shifts, and detailed accessories.

**Questions:**

- Modality Combination Justification : Why were emotion, identity, audio, and motion chosen specifically for joint modeling? Have you tested alternative subsets or fusion orders (e.g., removing emotion or treating identity differently)?
- Classifier-Free Guidance Role : In practice, most conditions other than motion appear to be applied through classifier-free guidance (Eq. 8). Does this setup provide genuine multi-modal fusion or merely weighted conditioning? Could this cause instability or overfitting on limited datasets?
- Emotion Control and Ablation : The supplementary results show only happy/sad emotion categories. Can the system handle subtle or continuous emotions (e.g., neutral, surprise)? A detailed ablation or user study on emotional control would strengthen the claims.
- Speech Injection at Late Layers : Why is the speech condition not injected at the final fusion layer, where fine-grained motion is refined? Have you tested injecting audio features at later stages to enhance lip-sync fidelity?
- Background Consistency : Some outputs exhibit background artifacts or unrealistic scenes. Is this due to the motion warping module or limitations of the rendering backbone (LivePortrait)? How might this be mitigated for higher realism?

**Details Of Ethics Concerns:**

The task involves generating human talking head videos, which inherently raises ethical considerations related to identity manipulation, consent, and potential misuse in deepfake applications. While the paper focuses on technical contributions, responsible research practices—such as data collection transparency, human subject consent, and safeguards against misuse—should be clearly discussed.

---

### Official Review · Reviewer_btbw · 2025-10-29

**Soundness:** 3
**Presentation:** 2
**Contribution:** 2
**Rating:** 4
**Confidence:** 4

**Summary:**

MoDA proposes a multi-modal talking-head framework, including a motion-generation DiT that fuses audio, identity and additional modalities (e.g., emotion) via joint-attention with RoPE, followed by a pre-trained LivePortrait-style renderer. The Motion Transformer uses coarse-to-fine (C2F) fusion strategy (four-stream -> two-stream -> single-stream) to progressively share parameters across modalities. Training uses rectified-flow loss (RF) with velocity loss and pre-trained Adaptive Lip-Motion Synchronization Expert (ALSE). The paper claims comparable or better sync quality with only 10 denoising steps, enabling real-time inference. Evaluation covers public datasets including HDTF and CelebV-HQ and in-the-wild cases.

**Strengths:**

- It is easy to read and the figures are easy to understand.
- The joint-attention formulation in simple and addresses cross-modal alignment.
- The coarse-to-fine (C2F) design is well-motivated and empirically supported. It reduces parameters dramatically, while improving sync and motion metrics.
- The 10-step rectified-flow variant appears effective.

**Weaknesses:**

1. Evaluation scope is narrow
Although public benchmarks provide established test splits, the paper evaluates on only 50 clips per benchmark and 20 in-the-wild examples. It is unclear why such a small subset as chosen despite the availability of full/standard test sets, and whether the 50 were truly random (or potentially cherry-picked). Please justify the subset size, describe the sampling protocol, and release the exact file indices to enable reproducibility. Also, the user study lacks detailed protocol (e.g., rater expertise, within-subject design, randomization.. )

2. Emotion control : metrics are insufficient
The paper claims controllable emotion, but the only "emotion" experiment (Table 7 in the Appendix) reports lip-sync metrics (Sync-C/Sync-D) and FVD, not emotion recognition/strength or consistency. Also, the user study only rates lip-sync, motion diversity and identity similarity. This makes it impossible to verify whether the intended emotion is actually expressed, how strongly, or how stably over time.

3. Metric concerns
There is metric fragmentation across tables, which prevents cross-table comparison:
- Table1 (main) : FVD, FID, F-SIM, Sync-C, Sync-D, Smo. (missing RTF)
- Table2 (in-the-wild) : F-SIM, Sync-C, Sync-D, RTF (missing FVD, FID, Smo)
- Table3 (ablations) : FID, FVD, Sync-C, Sync-D (missing F-Sim, RTF, Smo)
- Table7 (emotion) : Sync-C, Sync-D, FVD (missing F-Sim, RTF, FID, Smo)
- Table8 (architectural choices) : FVD, Sync-C, Sync-D (missing FID, F-Sim, Smo., RTF)
I wonder why do tables not share the same metrics.

4. ALSE design and potential leakage
The ALSE loss gating uses a hard cosine threshold \tau = 0.4, but the paper doesn't justify or analyze the threshold. Moreover, ALSE is trained on the same data distribution as MoDA, which can bias auxiliary signal toward the training distribution and inflate sync improvements. A calibration curve or cross-dataset validation is missing.

5. Motion smoothness loss is underspecified.
The paper introduces velocity loss to encourage temporal consistency, but does not define details. There is also a minor notation error in  line 268. Without targeted ablations, it is unclear whether the velocity loss improves naturalness or merely over-smoothes fast articulatory events, potentially hurting lip-sync.

6. Baseline coverage feels incomplete.
Several recent, widely discussed talking-head works (e.g., Hallo3, OmniAvatar, StableAvatar ..) are not included.

**Questions:**

1. ALSE threshold: Why \tau = 0.4? Please provide sensitivity and show if it improves or harms sync across unseen datasets.
2. ALSE independence : Since ALSE is trained on the same distribution, how do you avoid leakage? Can you train ALSE on disjoint speakers and report cross-dataset sync?
3. Velocity loss effect : Please analyze the effect of velocity loss.
4. Scale-up evaluation : evaluate on the whole test-splits of each benchmark.
5. Emotion metrics: Please add emotion accuracy or consistency metrics and include a user study section rating perceived emotion.
6. Missing metrics: Unify the metric set across all tables (at least FVD, FID, F-Sim, Sync-C, Sync-D, Smoothness).
7. SoTA model comparisons : Can you include more recent and widely-discussed works  (e.g., SadTalker, AniPortrait, Hallo3, OmniAvatar, StableAvatar ..)  and compare their performance?

---

### Official Review · Reviewer_fF2N · 2025-11-01

**Soundness:** 3
**Presentation:** 2
**Contribution:** 3
**Rating:** 4
**Confidence:** 2

**Summary:**

This paper presents a multi-modal diffusion framework that fuses identity, audio, and emotion information in a coarse-to-fine manner. The approach is conceptually clear and achieves solid quantitative and qualitative results, though the motivation for the specific fusion hierarchy could be explained in more depth.

**Strengths:**

- The paper is well organized and easy to follow.
- The proposed multi-stage fusion design is interesting and shows clear performance improvements.

**Weaknesses:**

1. The coarse-to-fine stage setup feels somewhat heuristic, and it is not entirely clear why identity, audio, and emotion are treated as “coarse” features.
2. While the proposed fusion strategy empirically improves performance, there is little analysis quantifying why each fusion step (e.g., merging audio with emotion and identity first) is optimal. The explanation remains largely intuitive, without supporting ablations or sensitivity analysis

**Questions:**

Could the authors clarify how the fusion stage order was determined and whether different fusion configurations were explored?
It would also be helpful to understand the reasoning behind treating certain modalities (e.g., identity, audio, emotion) as coarse features.

---

### Note · Authors · 2025-11-12

I have read and agree with the venue's withdrawal policy on behalf of myself and my co-authors.